# Mitochondrial Genome of *Fagus sylvatica* L. as a Source for Taxonomic Marker Development in the Fagales

**DOI:** 10.3390/plants9101274

**Published:** 2020-09-27

**Authors:** Malte Mader, Hilke Schroeder, Thomas Schott, Katrin Schöning-Stierand, Ana Paula Leite Montalvão, Heike Liesebach, Mirko Liesebach, Barbara Fussi, Birgit Kersten

**Affiliations:** 1Thünen Institute of Forest Genetics, D-22927 Grosshansdorf, Germany; malte.mader@thuenen.de (M.M.); hilke.schroeder@thuenen.de (H.S.); thomas.schott@thuenen.de (T.S.); stierand@zbh.uni-hamburg.de (K.S.-S.); ana.montalvao@thuenen.de (A.P.L.M.); heike.liesebach@thuenen.de (H.L.); mirko.liesebach@thuenen.de (M.L.); 2Center for Bioinformatics, Universität Hamburg, 20146 Hamburg, Germany; 3Bavarian Office for Forest Genetics, 83317 Teisendorf, Germany; barbara.fussi@awg.bayern.de

**Keywords:** mitochondrial genome, genome assembly, *Fagus*, Fagaceae, Fagales, molecular marker, mitochondrial marker, taxon assignment, CAPS marker, SNP

## Abstract

European beech, *Fagus sylvatica* L., is one of the most important and widespread deciduous tree species in Central Europe and is widely managed for its hard wood. The complete DNA sequence of the mitochondrial genome of *Fagus sylvatica* L. was assembled and annotated based on Illumina MiSeq reads and validated using long reads from nanopore MinION sequencing. The genome assembled into a single DNA sequence of 504,715 bp in length containing 58 genes with predicted function, including 35 protein-coding, 20 tRNA and three rRNA genes. Additionally, 23 putative protein-coding genes were predicted supported by RNA-Seq data. Aiming at the development of taxon-specific mitochondrial genetic markers, the tool SNPtax was developed and applied to select genic SNPs potentially specific for different taxa within the Fagales. Further validation of a small SNP set resulted in the development of four CAPS markers specific for *Fagus*, Fagaceae, or Fagales, respectively, when considering over 100 individuals from a total of 69 species of deciduous trees and conifers from up to 15 families included in the marker validation. The CAPS marker set is suitable to identify the genus *Fagus* in DNA samples from tree tissues or wood products, including wood composite products.

## 1. Introduction

*Fagus* is a genus of deciduous trees in the family Fagaceae (order Fagales), native to temperate Europe, Asia, and North America. The genus is divided into two subgenera, Engleriana and Fagus [1] and comprises 11 species (without ssp., var. and f.) [2]. As a naturally growing forest tree, European beech (*F. sylvatica* L.; subgenus Fagus) is among the most important and widespread tree species in Central Europe and is widely managed for its versatile hard wood. Because of the significance of the genus *Fagus*, genomic sequence resources are highly desired for this genus. 

Mishra et al. [3] published a 542 Mb nuclear draft genome sequence of an up to 300-year-old *F. sylvatica* individual (Bhaga) from an undisturbed stand in the Kellerwald-Edersee National Park in Central Germany. The assembly comprised 6451 scaffolds that are not yet assigned to any chromosomes. Complete chloroplast genomes are publicly available for *F. sylvatica* [4], *F. engleriana* [5], *F. crenata* [6], and *F. japonica* var. *multinervis* [7]. No complete mitochondrial genome sequence is available for any *Fagus* species to date [8]. In the entire order of the Fagales, only two assembled mitochondrial genome sequences are publicly available: one from *Quercus variabilis* (unverified sequence; GenBank MN199236) [9] and one from *Betula pendula* (GenBank LT855379.1) [10]; however, this sequence is not annotated so far.

Plant mitochondrial genomes are much larger than those of animals and highly variable in size [8,11,12,13,14,15,16]. They have low mutation rates, but have such high rearrangement rates that there is virtually no conservation of synteny [17,18,19,20,21,22]. The plasticity of plant mitochondrial genomes, leading to genome expansion, arises primarily from repeat sequences (including nontandem repeats of 50 bp and up), intron expansion, and incorporation of plastid and nuclear DNA [12,15,17,18,19,22,23,24,25,26,27,28]. The mitochondrial genome sequences of angiosperms generally have one or more pairs of large nontandem repeats (interspersed repeats) that can act as sites for inter- and intramolecular recombination, leading to multiple alternative arrangements (isoforms; including subgenomic forms) within a species [19]. Although plant mitochondrial genomes are often assembled and displayed as circular maps, plant mitochondrial DNA (mtDNA) does most likely not exist as one large circular DNA molecule but mostly as a complex and dynamic collection of linear DNA with combinations of smaller circular and branched DNA molecules [12,14,15,17,19,26,29,30,31,32,33,34].

DNA barcoding is an effective technique in molecular taxonomy. Sequences suggested to be useful in DNA barcoding include mtDNA (e.g., *cox1* for animals), chloroplast DNA (e.g., *rbcL*, *matK*, *trnL*-*trnF*, *ndhF*, and *atpB*), and nuclear DNA (ITS and house-keeping genes, e.g., GAPDH; [35,36,37,38,39] among others). Chloroplast and mitochondrial genes are preferred over nuclear genes because most of the genes lack introns, and they are generally haploid [40]. Furthermore, each cell has many chloroplasts and mitochondria, and each one can contain several copies of the respective genome [41,42,43]. Thus, when sample tissue is limited, the chloroplast and the mitochondrion offer relatively abundant sources of DNA. 

Advances in high-throughput sequencing (short read and single molecule long read sequencing) have promoted the assembly of complete DNA sequences of chloroplast and mitochondrial genomes and allowed for extending barcoding from single loci to whole genomes [44,45]. Especially, complete chloroplast genome (plastome) sequences provide valuable data sets to resolve complex evolutionary relationships in plastome phylogenies and improve resolution at lower taxonomic levels (e.g., [46,47]). Based on whole plastome alignments, genetic markers for species identification were developed in several studies (e.g., [44,48]). 

The lower nucleotide substitution rate in mitochondrial compared to chloroplast genomes of land plants [17,18,19,20,21,22] often provides not enough variation at the species level, although a few mitochondrial markers for potential species differentiation were developed (e.g., [49]). However, nucleotide variants in mitochondrial genomes may provide promising targets for the development of DNA markers that are specific for higher taxonomic levels, such as genus, family or order. 

European beech is often used in particle boards [50] and not always declared as it should be due to the European Timber Regulation that came into force in 2013 [51]. Although, species of the genus *Fagus* have not yet been added to the IUCN list [52] as “endangered” species (*F. longipetiolata* and *F. hayatae* classified as “vulnerable”), the specific ecosystems they form are in danger. Due to deforestation, old-growth forests are vanishing all over the world. Especially in Eastern Europe, the protected old forests consisting of oak, spruce, and beech have been declining during the last decades mostly because of human impact, i.e., mainly poor management practices and illegal logging of old valuable trees [53]. However, these forests play a key role for sustaining biodiversity and in climate change because they store a high amount of carbon for long time periods [53].

As a contribution to the preservation of valuable forest ecosystems, molecular markers can help to identify non-declared genera or species in different wood composite products. Identification of timber genera or species from solid wood products is much easier than from composite products as it may contain wood from many different genera or species. All the more important are molecular markers that are specific for a tree genus of interest ensuring a 100% classification probability (“golden markers”), and thus allowing for a doubtless identification of the related genus in wood composite products. To further increase the identification confidence in a tree genus, additional markers specific for higher taxonomic levels are highly desired. All potential taxon-specific markers should be validated in as many other tree species as possible (including species common in wood composite products). 

This study aimed at sequencing and annotating the complete mtDNA sequence of a representative *F. sylvatica* individual, which was used as a source for the development of molecular markers suitable to identify the genus *Fagus*. The developed markers (for the genus *Fagus*, the family Fagaceae, and the order Fagales) provide a useful tool set to verify the declaration “genus *Fagus*” using wood from tree tissues or wood products, including composite wood.

## 2. Results

### 2.1. Assembly and Annotation of the DNA Sequence of the Complete Mitochondrial Genome of F. sylvatica L.

The reference specimen (FASYL_29) sequenced in this study was selected from a set of genotyped beech trees from a provenance trial used also in a former study [54]. It originates from the German population Gransee/Brandenburg, which is located in the center of the natural distribution range of *F. sylvatica* (see also Section 4.1). 

The complete DNA sequence of the mitochondrial genome of FASYL_29 was assembled based on Illumina MiSeq reads (2 × 300 bp; 24×) and validated using long reads from nanopore MinION sequencing (3.2×; MiSeq and MinION reads are accessible at SRA PRJNA648273). For the validation of the mtDNA sequence, nanopore reads were mapped to the assembled sequence. The mapping result presented in Appendix A shows that the mtDNA sequence is completely covered by overlapping long reads from nanopore sequencing.

The mitochondrial genome of the *F. sylvatica* individual FASYL_29 was assembled into a single DNA sequence of a total length of 504,715 bp and an average GC content of 45.8% (GenBank MT446430; Figure 1). The assembly may best fit on a circular map (Figure 1). This display is not corresponding to the physical structure of the genome in vivo where it more likely exists in different conformations ([14,26,33] among others; see introduction).

Furthermore, the *F. sylvatica* mitochondrial genome contains 32 interspersed repeats greater than 50 bp (Appendix A) including two repeats greater than 300 bp: one inverted repeat of size 918 bp and one direct repeat of size 316 bp (Figure 1). One copy of the inverted repeat is located near an ancestral gene cluster consisting of the genes *rpl16*, *rps3*, and *rps19* [18] (Figure 1). In comparison, the mtDNA sequence of *Quercus variabilis* [9] contains 17 repeats greater than 50 bp and three repeats greater than 300 bp. The largest repeat is about 17.3 kbp in size. The mitochondrial genome of *Betula pendula* [10] contains 133 repeats greater than 50 bp and two repeats greater than 300 bp. The largest repeat is 474 bp long. Various fragments of the largest repeat of *Quercus variabilis* are included with high identity in the mtDNA sequence of *F. sylvatica* (52% of the repeat with about 97% identity included; Appendix A) and of *Betula pendula* (38% of the repeat with about 97% identity included; Appendix A). A comparison of all identified *F. sylvatica* repeats with the repeats in *Quercus variabilis* or *Betula pendula*, respectively, showed that several repeats are identical und many have high similarity (summary in Appendix A; BlastN results in Appendix A). One of the *F. sylvatica* repeats greater than 50 bp, namely repeat_11 (81 bp in length), is 100% identical to a *Quercus variabilis* repeat of the same length (repeat_12).

Chloroplast-like DNA sequence regions with more than 90% similarity to the *F. sylvatica* chloroplast genome sequence of the same individual, FASYL_29 (NC_041437.1) [4] account for about 0.72% of the FASYL_29 mitochondrial genome and are distributed among three distinct regions in the genome (region 1: 70,870–71,106 bp with 98.3% identity; region 2: 159,645–160,312 bp with 98.8% identity; region 3: 199,831–202,538 bp with 95.2% identity). These three regions were also featured by increased coverage (compared to all other regions) when mapping all available trimmed Illumina reads from DNA-sequencing of FASYL_29 to the related mitochondrial genome sequence (data not shown). The increased coverage is due to an unspecific mapping of chloroplast DNA-derived reads in addition to mtDNA-derived reads to these regions.

In total, 58 genes with predicted function were annotated, including 35 protein-coding genes, 20 tRNA, and 3 rRNA genes. The gene *mttb* is probably a pseudogene (MT446430). All of the known genes coding for subunits of proteins of the respiratory chain were identified including *sdh3* and *sdh4* (Figure 1). Several genes coding for small or large subunits of ribosomal proteins are missing, i.e., *rps2*, *rps7*, *rps10*, *rps11*, *rps13*, *rpl2*, and *rpl15*. The genes *nad1*, *nad2*, and *nad5* were predicted to be fragmented in five exons each, belonging to more than one distinct transcription unit (MT446430). The maturation of these genes requires cis- as well as trans-splicing events. For the following genes, more than one exon at one distinct transcription unit was predicted: *ccmFc*, *nad4*, *nad7*, *cox2*, and *rps3* (MT446430). The start codons of *nad1*, *nad4L*, and *cox1* are potentially created by RNA editing, as indicated by mappings of RNA-Seq data from two individuals of *F. sylvatica* (RNA-Seq reads accessible at SRA PRJNA648273) to the annotated mitochondrial genome sequence of *F. sylvatica* in the related regions (Appendix A). Additionally, 23 potentially protein-coding genes of unknown function were annotated based on ORF prediction from assembled RNA-Seq data (ORF1–23 in Figure 1). 

In Appendix A, the gene order of potential protein-coding genes annotated in the mitochondrial genome of *Liriodendron tulipifera* [18] was compared with that of *F. sylvatica* and *Quercus variabilis* [9] as another Fagales species (*Betula pendula* was not included in the global comparison because the mitochondrial genome has not been annotated so far). As expected, there is no conservation of synteny between *Liriodendron tulipifera* and *F. sylvatica*. Although *F. sylvatica* and *Quercus variabilis* are members of the same family (Fagaceae; in different subfamilies), no larger syntenic gene groups could be identified. However, several small collinear gene clusters inferred by Richardson et al. [18] as ancestral angiosperm gene clusters in *Liriodendron tulipifera* were also identified in the *F. sylvatica* mitochondrial genome (Appendix A). Ancestral gene clusters identified in *F. sylvatica* include among others the *sdh4*/*cox3*/*atp8*-cluster (cluster also in *Quercus variabilis*), the *atp4*/*nad4L*-cluster (not in *Quercus variabilis*), the *cob*/*rps15*/*rpl5*-cluster including *ccmFc* (physical separation of *ccmFc* from *cob*/*rps15* in *Quercus variabilis*), the *rps12*/*nad3*-cluster (also in *Quercus variabilis*), and the *rpl16*/*rps3*/*rps19*/*rpl2*-cluster without *rpl2* (*rpl2* is absent in *F. sylvatica*; cluster not in *Quercus variabilis*; Appendix A). 

In the comparison of the gene order (Appendix A), two gene clusters—not present in *Liriodendron tulipifera*—were identified in both *F. sylvatica* and *Quercus variabilis*: the clusters *ccmB*/*rpl10* and *cox1*/*sdh3*. The *ccmB*/*rpl10*-cluster was also identified in the mitochondrial genome of another Fagales member in the Betulaceae family, *Betula pendula* (Appendix A), whereas *cox1* and *sdh3* are physically separated in *Betula pendula (*draft annotation of *cox1* in Appendix A; *sdh3* annotation by Blast analysis). In contrast to *Betula pendula*, *F. sylvatica* and *Quercus variabilis* include the tRNA-gene *trnK* (UUU) upstream of the *ccmB/rpl10*-cluster (in a distance of about 2500 bp from *ccmB*; Appendix A). Interestingly, the *ccmB*/*rpl10*-cluster is not present in some non-Fagales members of the fabids analyzed, such as *Populus tremula* (NC_028096, family Malpighiales), *Vicia faba* (KC189947, Fabales; *rpl10* is not annotated), *Malus x domestica* (NC_018554, Rosales; *rpl10* is not annotated), and *Citrullus lanatus* (NC_014043, Cucurbitales; *rpl10* is not annotated). 

### 2.2. Identification of Potentially Taxon-Specific SNPs in Mitochondrial Genes

In this study, the software SNPtax was developed allowing for the identification of potentially taxon-specific SNPs in mitochondrial genes using GenBank files of related species and outgroup species of interest as input. The software identifies SNPs specific for a pre-defined taxon based on multiple alignments of genic sequences extracted from the GenBank files together with related taxonomic information. SNPtax is freely available on https://github.com/tsciow/SNPtax.

In the search for SNPs that are potentially specific for different taxa within the Fagales, the GenBank files of the following 13 tree species were used as an input for SNPtax analysis: three Fagales species, i.e., *F. sylvatica* (GenBank Acc. MT446430; mitochondrial genome assembled and annotated in this study), *Quercus variabilis* (MN199236), and *Betula pendula* (LT855379; draft annotation in this study, see Materials and Methods) as well as 10 non-Fagales species of deciduous trees and conifers, i.e., *Bombax ceiba* (NC_038052), *Eucalyptus grandis* (NC_040010), *Lagerstroemia indica* (NC_035616), *Populus alba* (NC_041085), *Populus davidiana* (NC_035157), *Populus tremula* (NC_028096), *Populus tremula* x *Populus alba* (NC_028329), *Liriodendron tulipifera* (NC_021152), *Ginkgo biloba* (NC_027976), and *Pinus taeda* (NC_039746).

In total, 18 protein-coding genes were identified that are annotated in all 13 species. These genes were considered for the identification of potential taxon-specific SNPs. Only SNPs in conserved regions were taken into account. This way, we could identify 30 SNPs in 11 genes potentially specific for *F. sylvatica*, 29 SNPs in nine genes for Fagaceae (potential Fagaceae-specific allele occurred only in the sequences of *F. sylvatica* and *Quercus variabilis*; see above), and 27 SNPs in nine genes for Fagales (specific allele only in *F. sylvatica, Quercus variabilis* and *Betula pendula*). All SNPs potentially specific for *F. sylvatica*, Fagaceae, and Fagales are summarized in Appendix A. The SNPs are located in 13 different mitochondrial genes. 

### 2.3. Development of Selected CAPS Markers and Further Validation of Their Taxon Specificity

From potentially taxon-specific SNPs (Appendix A), some SNPs were selected for the development of cleaved amplified polymorphic site (CAPS) markers which fulfill the following criteria: (i) a SNP is located in a recognition site of a restriction enzyme; (ii) a SNP allele of the target taxon is part of the recognition site sequence, i.e., amplicon of the target taxon will be cut by the restriction enzyme whereas the others will not; (iii) a SNP is not located close to one of the gene ends (design of flanking primers is possible within the gene).

Specific primers were designed in the flanking regions of selected SNPs considering that the amplicon size should not exceed 200 bp because mtDNA extracted from processed wood products may be highly degraded. Moreover, primer sequences should match the respective genic region in all 13 species included in the genic alignments (see above) with a perfect match to Fagales species (as far as possible) and with a perfect match or at most one to two mismatches to the other sequences. 

Selected CAPS markers were then pre-validated using DNA samples of different deciduous tree and conifer species. As an example, Figure 2 presents the pre-validation of the potential *Fagus*-specific CAPS marker 3_Fagus_*ccmFc* (marker description in Table 1). Only the five *Fagus* individuals from five different *Fagus* species included in the pre-validation provided the pattern of the digested PCR product with two fragments of 108 and 45 bp each, whereas all individuals from the other genera of deciduous trees or conifers showed the band of the non-digested PCR product of 153 bp (Figure 2).

Only CAPS markers that successfully passed the pre-validation were subjected to extended validation as detailed below (see also Section 4.10). Table 1 summarizes the features of four successfully validated mitochondrial CAPS markers, and Figure 3 compares the related digestion patterns of these markers in one individual each of *F. sylvatica* (Fagaceae, Fagales), *Quercus robur* (Fagaceae, Fagales), *Betula pendula* (Betulacea, Fagales), and *Populus tremula* (Salicaceae, Malpighiales) as a non-Fagales species.

In the entire validation (pre-validation and extended validation), each CAPS marker (Table 1; Figure 3) was successfully validated with around 100 individuals of 59–63 tree species that belong to 14–15 families (including four Fagales families) in nine to ten orders in total (see also 4.10). All species within the entire validation, the numbers of individuals per species, and details about the origin of the individuals are summarized in Appendix A. The species list contains—among others—species that are known to be potentially included in wood composite products. 

Besides, the two *Fagus*-specific CAPS markers—3_Fagus_*matR* and 3_Fagus_*ccmFc* (Table 1)—were successfully validated in 25 additional *F. sylvatica*, five additional *F. orientalis*, four additional *F. grandifolia* and *F. engleriana* each, and two additional *F. crenata* individuals (Appendix A). Although these two *Fagus-*specific markers were originally selected based on potentially *F. sylvatica*-specific SNPs (identified in the genic alignments described above including *F. sylvatica* as the only *Fagus* species), they turned out to be specific for the genus *Fagus* in the validation. 

## 3. Discussion

In this study, we present the complete DNA sequence of the mitochondrial genome of *F. sylvatica* L. (Figure 1; GenBank MT446430), a deciduous tree species in the Fagaceae family. This sequence is the first complete mitochondrial genome sequence for the genus *Fagus* and the third one for the order Fagales, together with mtDNA sequences of *Quercus variabilis* (GenBank MN199236; unverified) [9] and *Betula pendula* (LT855379.1; not annotated) [10]. The mitochondrial genome sequence of the *F. sylvatica* individual FASYL_29 represents the second extranuclear genome sequence of this individual, in addition to the already published chloroplast genome sequence (NC_041437.1) [4].

Although short reads (Illumina MiSeq reads of 2 × 300 bp) were used in this study, the complete mitochondrial genome sequence could be combined from two large contigs of the initial assembly. The assembled sequence has been successfully validated by mapping of nanopore MinION long reads (Appendix A). Because short reads encounter numerous difficulties due to low-complexity homopolymeric sequence characteristics and the potential presence of large repeat regions in mitochondrial genomes [12,17,19,22,26,42], long read sequencing is increasingly applied (often in addition to short read sequencing) for subsequent assemblies of mitochondrial genome sequences (e.g., [57,58,59]).

The identified size of the *F. sylvatica* mitochondrial genome assembly of 504,715 bp is between the size of the mitochondrial genomes of *Quercus variabilis*—another member of the Fagaceae family—of 412,886 bp (GenBank MN199236) [9] and *Betula pendula* (Betulaceae) of 581,505 bp (LT855379.1) [10]. 

Mitochondrial genomes of flowering plants are well known for their large size, fluid genome structure, and variable coding-gene set often due to horizontal gene transfer; e.g., chloroplast and nuclear sequences have been found in mitochondrial genomes or vice versa [11,12,13,18,19,22,23,24,25,26,27,28,56]. These are ongoing processes in plants. Most of the transfers in angiosperms involve ribosomal protein genes [60]. Thus, it is not unexpected that seven genes (*rps2*, *rps7*, *rps10*, *rps11*, *rps13*, *rpl2*, *rpl15*) of the ribosomal genes belonging to the ancestral gene content of the mitochondrial genome of flowering plants [18] were not annotated in *F. sylvatica* (Figure 1). Five of these seven genes (with the exception of *rps10* and *rpl2*) are also missing in the mitochondrial genome of *Quercus variabilis* (MN199236) [9]. The missing genes *rps2* and *rps11* are also lacking in the mtDNA of *Ricinus communis* [61], *Hevea brasiliensis* [62], and *Populus tremula* [44,63] among others. The absence of the *rps13* gene from mitochondrial genomes has been shown for many members of the rosids subclass [60] including *F. sylvatica* (in this study). The ribosomal gene *rps10* missing in *F. sylvatica* is also missing e.g., in *Populus tremula* [44] and *Hevea brasiliensis* [62], but present in *Ricinus communis* [61]. A loss of *rps7* was also reported in ancestors of the Fabaceae family and of *rpl2* in some Fabaceae species [64]. Although the *rpl5* gene is lacking from many of the sequenced plant mitochondrial genomes [65], it is annotated in *F. sylvatica* (MT446430). The two respiratory genes—*sdh3* and *sdh4* (encoding subunits 3 and 4 of succinate dehydrogenase)—that have been reported to be lost from the mitochondrial genome of various angiosperms [66], were annotated in *F. sylvatica* (MT446430). 

Plant mitochondrial genomes have abundant interspersed repeats [12,17,19,22,26,42] often including pairs of large repeats which cause isomerization of the genome by recombination, and numerous repeats of up to several hundred base pairs that recombine only when the genome is stressed by DNA damaging agents or mutations in DNA repair pathway genes [19]. In general, the largest repeats within a species (in angiosperms often longer than about 1 kb) have been found to recombine constitutively, leading to isomerization [19]. The longest interspersed repeat in the mtDNA of *F. sylvatica* is about 1 kb (918 bp in size; Appendix A) and may be responsible for isomerization. Whereas the longest repeat in the mtDNA of *Quercus variabilis* (another Fagaceae member) is 17.3 kb in size, the longest *Betula pendula* repeat is only 474 bp (Appendix A). By comparing mtDNA sequences of 72 angiosperm species, Wynn and Christensen [19] found that only a part of the species (43%) shows repeats above 10 kb.

The dynamic nature of mitochondrial genomes in the Fagales is also reflected by a gene order comparison between *F. sylvatica* and *Quercus variabilis* (Appendix A) which both are members of the Fagaceae family. Although some small collinear gene clusters inferred by Richardson et al. [18] as ancestral angiosperm gene clusters in *Liriodendron tulipifera* were also identified in the mitochondrial genomes of *F. sylvatica* and *Quercus variabilis*, no larger syntenic gene groups could be identified. Interestingly, two common gene clusters—not present in *Liriodendron tulipifera*—were identified in *F. sylvatica* and *Quercus variabilis*: the clusters *ccmB*/*rpl10* and *cox1*/*sdh3*. Whether the *ccmB*/*rpl10*-cluster, which was also identified in *Betula pendula* (Appendix A), is a common cluster of all Fagales remains an open question for future research. 

Plant mitochondria employ distinct and complex RNA metabolic mechanisms including RNA editing, splicing of group I and group II introns, maturation of transcript ends, and RNA degradation (reviewed in [34]). RNA editing (in the form of C-U base transitions) is a post-transcriptional process which is highly prevalent in mitochondria and chloroplasts of land plants [67]. Numerous C→U conversions (and in some plants also U→C) alter the coding sequences of many transcripts of the organellar genomes, while e.g., eliminating premature stop codons or creating AUG start sites, as also shown in this study for the start sites of *nad4L, cox1*, and *nad1* (Appendix A). The start codon of *cox1* is also generated by RNA editing in other land plants, e.g., *Liriodendron tulipifera*, *Nelumbo nucifera*, *Nicotiana tabacum* [68], and *Solanum tuberosum* [69]. The start codons of *nad1* and *nad4L* are also created by RNA editing in *Allium cepa*, *Cucumis sativus*, *Glycine max*, *Gossypium hirsutum*, *Liriodendron tulipifera*, *Nelumbo nucifera*, *Oryza sativa*, *Phoenix dactylifera*, and *Zea mays* [68], among others. In general, non-synonymous RNA editing sites were shown to be particularly highly conserved across different plant species ([68,70] among others).

Aiming at the development of mitochondrial genetic markers suitable to identify *Fagus* species from potential mixtures of different tree species in wood composite products, we sought SNPs specific for *Fagus*, Fagaceae, and Fagales in this study. In contrast to other studies that focused on classical plant barcoding regions (e.g., [71,72,73]), we followed a strategy similar to super-barcoding [45], however, not considering the entire mitochondrial genome, but including all mitochondrial genes common in tree species used for marker development. Because of the highly dynamic structure of mitochondrial genomes of angiosperms, alignments of complete mitochondrial genome sequences make sense only in very closely related individuals. Recombination activities involving repeated sequences may generate subgenomic forms and extensive structural variation of angiosperm mitochondrial genomes even within the same species [11,12,14,15,17,19,26,29,30,31,32,33,34].

The development of the SNPtax tool allowed us to select SNP markers potentially specific for different pre-defined taxa based on alignments of DNA sequences of mitochondrial genes (also considering intron-containing genes but excluding trans-spliced genes). The screen for taxon-specific SNPs in conserved genic regions allows considering a broad taxonomic range during the initial SNP selection and also during marker validation because primers can be designed that amplify the region of interest in tree species of various families. The developed CAPS markers (Table 1, Figure 3) are specific for the taxa *Fagus*, Fagaceae, or Fagales, respectively, when considering the tree individuals and related species (59–63 species from about 15 families and 10 orders) included in the entire validation for each marker (see also Appendix A). All CAPS markers (Table 1) are located in exonic regions of the related genes with the exception of the marker 4_Fagaceae_*nad7* that is based on a Fagaceae-specific SNP in intron 2 of the *nad7* gene. An intron of the *nad7* gene (fourth intron region) was also considered in a study aiming at the identification of medicinal plants [74].

Further validation of the CAPS markers developed in this study is necessary to prove their taxon-specificity also in extended sets of tree individuals from various species, especially if they should be applied for taxon identification against a broader species background than the potential species spectrum of wood composite products. In particular, the two potentially *Fagus*-specific CAPS markers should be further validated with other *Fagus* species besides the five *Fagus* species included in this study. 

Molecular markers for taxon assignment within the Fagaceae were also developed in previous studies. For species identification among common tree species of the Alps, Brunner et al. [75] developed CAPS markers based on SNPs in the intron of the plastid gene *trnL* (UAA). One of the markers allows for differentiating *F. sylvatica* from 21 other tree species tested in this study. Because no other *Fagus* species were analyzed, it is unclear if the marker is specific only for *F. sylvatica* or also for other *Fagus* species [75]. Unfortunately, an application of this marker with highly degraded DNA from processed wood products is probably not feasible (amplicon size is too large for this purpose). In another study, microsatellite primers were developed for the endangered beech tree species, *Fagus hayatae* [76]. Recently, a set of 58 SNPs has been selected from coding regions and applied for species discrimination among European white oaks [77]. Different types of molecular markers for DNA profiling of *Quercus* spp. or *Quercus* species groups were developed in other studies, e.g., based on plastid SNPs and InDels [78], short tandem repeat loci [79], or inter-primer binding sites [80].

Recent advances in real-time nanopore sequencing could pave the way to species identification using genome scale data in the future as shown in a field-based study of closely-related *Arabidopsis* species [81]. 

## 4. Material and Methods

### 4.1. Plant Material

The reference *F. sylvatica* specimen for DNA sequencing (FASYL_29) was selected for sequencing from a set of genotyped beech trees from a former study [54]. It originates from the German population Gransee/Brandenburg, located in the center of the natural distribution range (53°00′ N, 13°10′ O, 70 m a.s.l.) and was sampled in a progeny test. The parental population is an approved seed stand at the age of about 200 years. The selected individual is a good genetic representative of Central Europe with highest similarities to German, Czech, and Austrian beech populations.

For RNA sequencing, buds of three year old seedlings (HE1 and HE2) grown in a nursery were collected in winter. The seedlings originate from a seed lot harvested in a natural *F. sylvatica* population that is located close to Hengstberg in the Fichtel Mountains in Germany (geo-coordinates of the population: 50°07′55′′; 12°11′18′′).

For validation of the developed CAPS markers, leaves or buds from 145 trees of 69 different species and various geographic origins were used. A part of this plant material was sampled in the Arboretum of the Thünen Institute of Forest Genetics in Großhansdorf. Additionally, DNA from specimens of *F. sylvatica* was provided by the projects “GenMon” and “Herkunft Europa” (subproject/project conducted at the Thünen Institute of Forest Genetics). The *Quercus* specimens were provided by the previous project “DBU Weisseiche” and by North American, Russian, and Korean institutions. Different Botanical Gardens provided us also with specimens of other species (Appendix A). For the sampling strategy, we always tried to obtain material from different regions of the natural distribution areas of the tree species. Thus, e.g., especially for *F. sylvatica*, individuals from all over Europe were used (Appendix A). The use of a broad range of species within the orders and families was particularly important to select “golden” markers with high specificity based on validation.

### 4.2. DNA Sequencing of F. sylvatica L.

Dormant buds of the *F. sylvatica* individual FASYL_29 were sampled, green tissues were dissected and DNA was extracted for short read sequencing following a slightly modified ATMAB protocol based on the protocol of Dumolin et al. [82]. Standard genomic library preparation and 300 bp paired-end sequencing was performed on Illumina MiSeq at 24× coverage (GATC Biotech AG, Konstanz, Germany). 

For long read sequencing, DNA was extracted using a combination of a lysis buffer, Sera-Mag Speed beads, and a purification step, adapted from [83]. Briefly, the outer layer of leaf buds was removed; then, they were cut into pieces (to facilitate grinding), collected into 2 mL Eppendorf tubes, and frozen in liquid nitrogen (10 buds in total; two buds per tube). Samples were ground using a Retsch Mill (Retsch MM300, Haan, Germany), with two stainless steel beads (5 mm) per tube, at a speed of 25 Hz for 35 s. DNA lysis was performed by adding 700 µL lysis buffer (1% PVP 40 (*w*/*v*), 1% PVP 10 (*w*/*v*), 500 mM NaCl, 100 mM Tris-HCl pH 8.0, 50 mM EDTA, 5 mM DTT, 1.25% SDS (*w*/*v*), and 1% Sodium metabisulfite (*w*/*v*)) to each of the samples, which were then mixed gently by flicking and incubated at 64 °C for 30 min. Following that, 1 µL RNase A (10 mg/mL) per 1 mL lysis buffer was added, and the samples were incubated at 37 °C for 50 min at 400 rpm on a thermomixer. After the first 20 min, 10 µL Proteinase K (800 units/mL) was added to each sample. Once the incubation time ended, samples were left on ice for 2 min to cool down, and 0.3 volume 5 M potassium acetate pH 7.5 was added. Samples were manually mixed by inversion 20 times, slowly, then centrifuged at 8000× *g* for 12 min at 4 °C.

For DNA size selection, the supernatant was transferred to clean 1.5 mL LoBind Eppendorf tubes and 0.8 V of a homogenized Sera-Mag Speed beads solution (10 mM Tris-HCl, 1 mM EDTA pH 8.0, 1.6 M NaCl, 11% PEG 8000, 0.4% beads (*v*/*v*)) was added, and the tubes were mixed gently by flicking. The samples were placed on a rotor for 10 min, briefly centrifuged, and placed on a magnet. Once the beads were on the back of the tubes and the solution became clear, the supernatant was discarded, and the beads were washed twice with 1 mL freshly prepared 70% ethanol. After the last ethanol wash, the tubes were removed from the magnet and briefly centrifuged. After placing them back on the magnet, the last drops of ethanol were pipetted off. The beads were air dried for 30 s; then, the tube was removed from the magnet, and 50 µL pre-heated (50 °C) 10 mM Tris-HCl pH 8.0 was added for elution. The tubes were flicked to resuspend the beads then incubated for 10 min at room temperature. Finally, the tubes were placed back on the magnet, and once the solution was clear, it was transferred to fresh tubes.

The DNA purification step was performed using chloroform:isoamylalcohol (24:1). Since the extraction was performed using multiple tubes, the eluted DNA from each tube (~80 µL) was pipetted into a single tube, comprising a total of 400 µL. Then, one volume of chloroform:isoamylalcohol was added, and it was mixed by inversion for 5 min on a rotor. After that, the tube was centrifuged at 5000× *g* for 2 min at room temperature, and the upper phase was transferred to a fresh tube. The chloroform:isoamylalcohol step was repeated, and after that, a 0.1 volume of 100% cold ethanol was added, and the sample was centrifuged at 5000× *g* for 2 min at room temperature. The pellet was washed with 70% ethanol and resuspended in 50 µL 10 mM Tris-HCl pH 8.0 for 2 h at room temperature. The sample was stored at 4 °C until library preparation. 

The sequencing library was prepared using the Ligation Sequencing Kit (SQK-LSK109) following the manufacturer’s instructions (Oxford Nanopore Technologies, Oxford, UK), with 2 µg DNA as input. The R9.4.1 MinION Flow Cell was primed with the Flow Cell Priming Kit (EXP-FLP002). Sequencing was performed on a MinION Mk1B device (MIN-101B) connected to a MinIT computer (MNT-001). We used the MinIT software version 19.01.1, and further basecalling was performed using guppy v3.2.2.

### 4.3. RNA Sequencing of Two F. sylvatica L. Individuals

Bud samples from three-year-old *F. sylvatica* seedlings (HE1 and HE2) growing in a nursery were collected and immediately put on dry ice. RNA extraction and all further steps were performed by IGA Technology Service in Udine (Udine, Italy). Libraries were prepared using Illumina TruSeq mRNA-seq Kit. Clusters were generated on a flowcell by cBot and sequenced on a HiSeq2000 by using standard Illumina sequencing workflow. 

### 4.4. Assembly and Scaffolding of the Illumina MiSeq Reads from DNA Sequencing

Reads were trimmed with Trimmomatic version 0.36 [84] and assembled using the CLC Genomics Workbench (CLC-GWB) Version 10.1.1 (CLC-bio, a Qiagen company; Aarhus, Denmark), (length fraction = 0.9, similarity fraction = 0.95, map reads back to contigs, word size = 45). Two out of 279,689 contigs were identified as mitochondrial sequences based on high mapping coverage and comparison against the NCBI nucleotide collection database. Blastn was used to join the mitochondrial contigs by finding similarities between the sequence endings in order to create the complete DNA sequence of the mitochondrial genome. To verify the assembled sequence, we used the tool “ROUSFinde1_1”, described by Wynn and Christensen [19], with default parameters to identify repeat structures in the mitochondrial sequence of *F. sylvatica* (MT446430). Accordingly, we identified repeat structures in *Quercus variabilis* (MN199236) and *Betula pendula* (LT855379.1) for the comparison of repeats between these three mitochondrial genomes (Appendix A).

### 4.5. Mapping of Nanopore MinION Reads to the DNA Sequence of the Mitochondrial Genome of F. sylvatica

MinION reads were error-corrected using the “Correct PacBio reads 1.1.” tool of the Genome Finishing Module of CLC-GWB v12.0 (coverage percentage of reads to correct = 40). Corrected reads of a length above 10,000 bp were mapped onto the final assembly of the mitochondrial genome of *F. sylvatica* by the “Map reads to contigs 1.3” tool of CLC-GWB v20.04 using default parameters with increased length and similarity fraction (length fraction = 0.98; similarity fraction = 0.95). The created stand-alone read mapping is presented in Appendix A.

### 4.6. Assembly of the RNA-Seq Data

RNA-Seq data from the samples HE1 and HE2 were used for the annotation of additional ORFs. Reads were trimmed with Trimmomatic version 0.36 [84]. The combined trimmed reads of HE1 and HE2 were mapped with the STAR RNA-Seq aligner to the mitochondrial reference sequence of *F. sylvatica*. This mapping was used for a reference-guided assembly of the RNA-Seq data with the software Trinity version 2.8.5 [85] which resulted in 229 mitochondrial transcript contigs.

### 4.7. Annotation of the DNA Sequence of the Mitochondrial Genome of F. sylvatica

Structural and functional annotation was performed using the GeSeq server [86] with default settings and NC_028096 (mtDNA sequence of *Populus tremula*) set as reference. Using the Sequin tool v13.05 [87], these draft annotations were corrected where necessary, guided by alignments to other well-characterized eudicot mtDNA sequences including those of *Arabidopsis thaliana* (NC_037304.1) [88], *Cucurbita pepo* (NC_014050.1) [65], and *Quercus variabilis* (MN199236) [9]. Additional ORFs including the ORF of the gene *rpl10* were identified by ORF prediction in transcript contigs assembled from RNA-Seq data of two individuals (see Section 4.6) using the “Find open reading frames”—tool of CLC Genomics Workbench version 20.0.3 with the following parameters: Start codons = AUG; Stop codon included in annotation = Yes; Annotate sequences = Yes; Both strands = Yes; Open-ended sequence = No; Genetic code = 1 Standard; Minimum length (codons) = 100. In total, 90 ORFs with a minimal length of 300 bp (including start and stop codons) were identified and gave rise to 24 additional annotations (ORF 1–23 and *rpl10*), which were manually added to the Seqin file (annotated as “hypothetical proteins”). Nested ORFs were not considered. If BlastP analysis of the amino acid sequence of a related ORF versus NCBI non-redundant proteins provided hits with at least 80% query coverage and 90% identity to any known protein sequence; then, this information was included into the protein description (as a comment) in the Seqin file.

To compare and visualize the gene order between different mitochondrial genomes, the web version of the software tool geneCo (gene Comparison) [89,90] was applied (using the “map comparison” option with default parameters, but displaying names of all genes; considering only potential protein-coding genes and excluding trans-spliced genes).

### 4.8. Multiple Alignments of Mitochondrial Gene Sequences and Selection of SNPs Specific for Pre-Defined Taxa

SNPs specific for selected taxa were identified based on the newly developed software SNPtax which uses a set of custom scripts based on BioPerl [91] and PRANK v.170427 [92] as alignment tool. By extracting the taxonomic information together with the genic sequences from different mitochondrial genomes in GenBank format and then generating multiple alignments for each gene—at every position in the alignment—the taxon can be determined for which a certain base is characteristic. The resulting output can then be screened for selected taxa of interest to retrieve taxon-specific SNPs. 

In total, the complete mitochondrial genome sequence of 13 tree species (see Section 2.2) were included in the analysis presented in this study. Because the *Betula pendula* genome (LT855379) was only available without annotation, the sequence was submitted to the GeSeq server [86,93] for gene calling and functional annotation with NC_028096 as a reference. The resulting GenBank file was used as input for the SNPtax analysis beside the GenBank files of the other 12 species. 

### 4.9. DNA Preparation, PCR Amplification, and Agarose Gel Electrophoresis for CAPS Markers

Either buds, leaves, or cambium were prepared for extraction of total DNA following a slightly modified ATMAB protocol according to Dumolin et al. [82]. The type of tissue was dependent on what was provided as reference material used in this study. 

PCR amplification was performed in 20 µL volume with 20 ng DNA. For all markers shown in Table 1, the reaction mixture contains 1× BD Buffer, 2 mM MgCl_2_, 200 µM each dNTP, 1xDMSO (NEB), 0.2 µM of each Primer, and 1 unit Taq Polymerase (DCS Pol, DNA Cloning Service, Hamburg), except for marker 5_Fagales_*mat*R (1.5 unit Taq polymerase was used). The PCR was performed with the following program: 94 °C for 4 min, followed by 35 cycles with 94 °C for 45 s, 58 °C for 1 min, 72 °C for 1 min, and additional 5 min at 72 °C final elongation at the end.

For the restriction analyses for each marker, 10 µL PCR product was used in a volume of 20 µL. The restriction analyses (in deviation from the manufacturer’s protocols) were performed as follows: the PCR product amplified with the marker 5_Fagales_*matR* was digested with 4 units of the enzyme *Nci*I for 8 h at 37 °C and 20 min of inactivation at 80 °C—the same conditions were used for 3_Fagus_*matR* using the enzyme *Bst*XI; 4_Fagaceae_*nad7* was digested with the enzyme *Sfc*I with the same conditions as the former; 3_Fagus_*ccmFc* was digested with 5 units of the enzyme *Bsm*BI for 8 h at 55 °C and 20 min of inactivation at 80 °C. Restriction products were visualized relative to a 50 bp ladder (Life technologies, Germany, Martinsried) using a 1.5% agarose gel stained with Roti-Safe Gelstain (Carl Roth, Germany, Karlsruhe).

### 4.10. Validation of the CAPS Markers

The Fagales-specific marker 5_Fagales_*matR* was tested for “Fagales-specificity” using four families with ten genera, 41 species, and 75 individuals within the order Fagales, and additionally, nine other orders including 11 families, 17 genera, and 25 species with one individual each (Appendix A). For validation of the Fagaceae-specificity of the marker 4_Fagaceae_*nad7*, three genera, and 23 species with 57 individuals from the family Fagaceae, and three further families within the order Fagales using seven genera and 15 species were used. Additionally, outside the Fagales, nine orders including 11 families, 17 genera, and 26 species and individuals were tested for validation of this family-specific marker. For validation of both *Fagus*-specific markers—3_Fagus_*matR* and 3_Fagus_*ccmFc*—58 or 60 *Fagus* individuals were used, respectively. Outside from the genus *Fagus*, further 76/79 individuals from 55/57 species in 25 genera, 13 families, and eight orders were tested (Appendix A).

## Figures and Tables

**Figure 1 plants-09-01274-f001:**
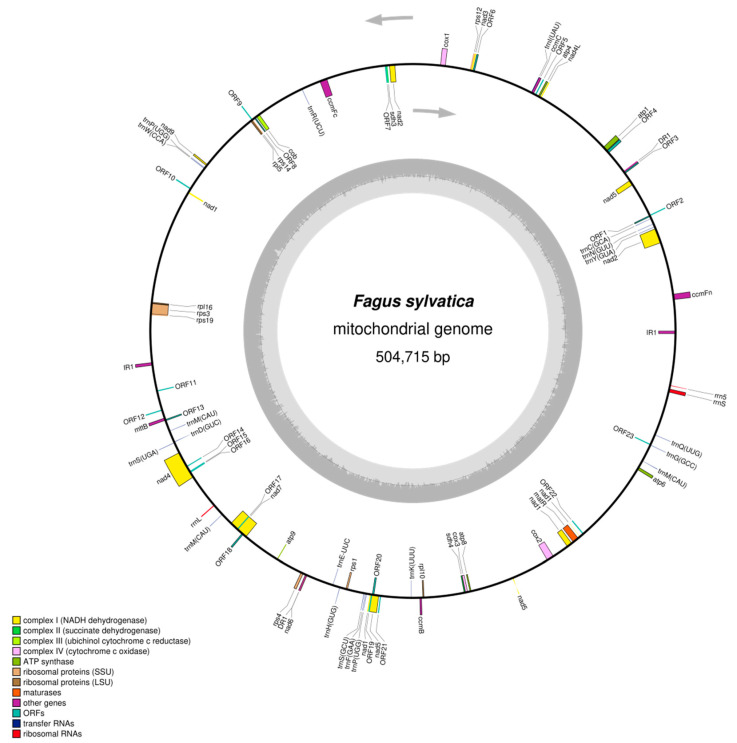
Circular graphical display of the assembled mitochondrial genome sequence of the *F. sylvatica* individual FASYL_29 (GenBank MT446430). The display does not correspond to the physical structure of the genome in vivo where it more likely exists in different conformations (see main text). Pairs of interspersed direct (DR) and inverted (IR) repeats longer than 300 bp and with ≥99% sequence identity are numbered (one pair each). In addition to protein-coding and structural RNA genes of predicted function, 23 potential CDS regions (indicated as “ORFs”) of unknown function with support from RNA-Seq data were predicted and mapped to the genome sequence. The grey arrows indicate the direction of transcription of the two DNA strands. A GC content graph is depicted within the inner circle. The circle inside the GC content graph marks the 50% threshold. The map was created using OrganellarGenomeDraw [55,56].

**Figure 2 plants-09-01274-f002:**
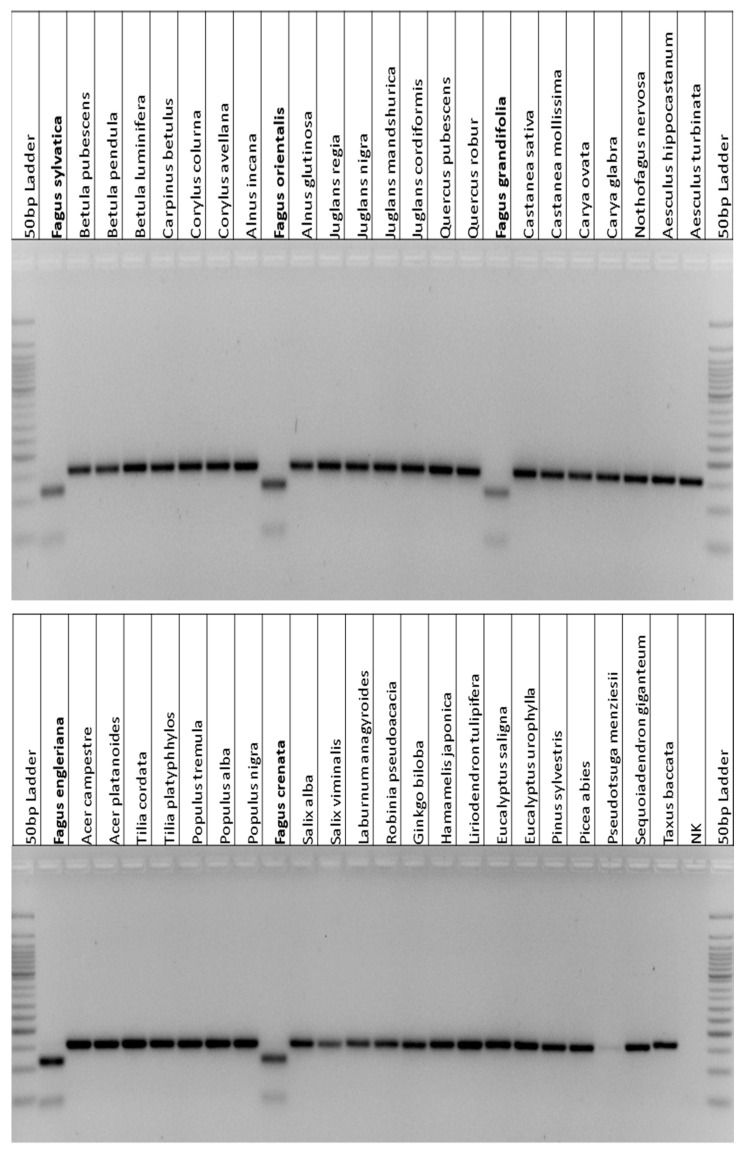
Pre-validation of the *Fagus*-specific mitochondrial CAPS marker 3_Fagus_*ccmFc* (marker description in Table 1) with DNA samples of *F. sylvatica*, and four other *Fagus* species as well as 41 non-*Fagus* species comprising 35 species of deciduous trees, 5 conifer species, and *Ginkgo biloba*. After digestion, PCR fragments were separated on a 1.5% agarose gel. Related primer sequences are provided in Appendix A. Extended validation of CAPS markers was performed with much more individuals (see main text, Section 4.10, and Appendix A).

**Figure 3 plants-09-01274-f003:**
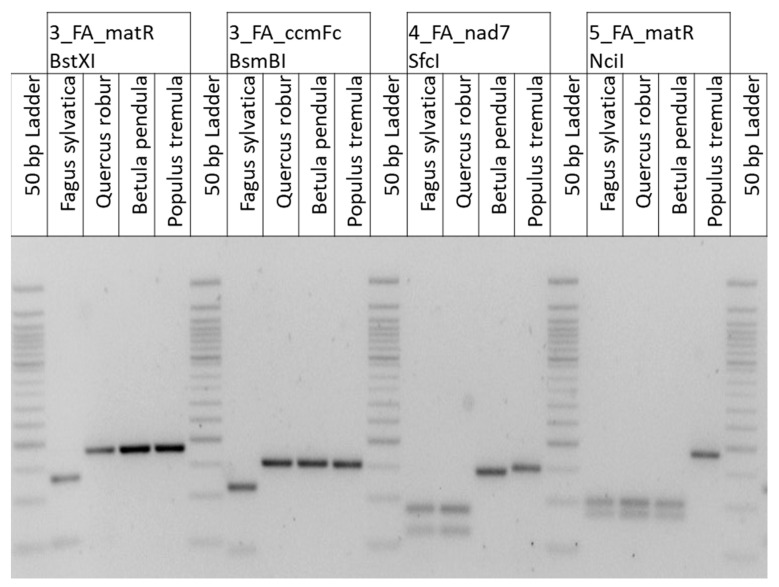
Representative restriction patterns of the CAPS markers (Table 1) analyzed in individuals of three Fagales and one non-Fagales species each. After digestion, PCR fragments were separated on a 1.5% agarose gel.

**Table 1 plants-09-01274-t001:** Mitochondrial CAPS markers specific * for genus *Fagus*, family Fagaceae, or order Fagales.

Marker Name	Taxon Specificity *	SNP Position (in bp)	Target Allele/ Alternative Alleles	PCR Amplification in Deciduous Trees (D)/ Conifers (C)	Restriction Enzyme	**Fragment Sizes** **(Target/Others)**
3_Fagus_*matR*	*Fagus*	1530	G/ C,T,A	D	*Bst*XI	53,129/ 182
3_Fagus_*ccmFc*	*Fagus*	1862	T/ C	D and C	*Bsm*BI	108,45/ 153
4_Fagaceae_*nad7*	Fagaceae	1459	A/ G	D	*Sfc*I	5786/ 143
5_Fagales_*matR*	Fagales	261	G/ A	D and C	*Nci*I	8897/ >=185

The name of the gene where the SNP is included is part of the marker name, e.g., “*matR*”. SNP positions are related to the nucleotide position in the DNA sequence of the respective gene in *Populus tremula* (GenBank KT337313.1) [44] used as reference. The “target” is the taxon given in column “taxon specificity”. Related primer sequences are provided in Appendix A. *, The term “taxon specificity” in this context means the specificity of the related marker compared to all individuals of different deciduous tree species and conifer species included in the marker development (see above) and validation in this study (Appendix A and Section 4.10).

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
