# Peer review of "Mitochondrial Genome of Fagus sylvatica L. as a Source for Taxonomic Marker Development in the Fagales"

_plants, 2020, doi:10.3390/plants9101274_

Round 1

Reviewer 1 Report

The work seemed to be quite useful, it is not only the description of new mitochondrial genomes for the plant but held the search of molecular markers for the identification of different plant taxa. That type of work is very important as the number of suitable markers for plant organisms is very small. I have a small comment not to the work itself, but to the documentation for the SNPtax software. The authors must clearly describe the input and output file formats. Without this information, it is very difficult to use the software provided.

Reviewer 2 Report

The manuscript entitled ”Mitochondrial genome of Fagus sylvatica L. as a source for taxonomic marker development in the Fagales”, written by   Mader et al. appears heterogenous with two parts that need to be completed :

- a first part which describes the use of a state of the art sequencing that results in a well annotated mt genome  but that could be better  described and compared with the 2 other Fagales mt genomes that are available. In addition, the circular structure is not discussed, while it has been put into question in the last decade (see Kozik A, Rowan BA, Lavelle D, Berke L, Schranz ME, Michelmore RW, et al. (2019) The alternative reality of plant mitochondrial DNA: One ring does not rule them all. PLoS Genet 15(8): e1008373). The figure S1 needs also a legend.

- a second part, that is the development of “old fashioned” markers specific to Fagus based on a multialignement of a large number of tree species. This part raises several questions:

Why did the authors decide to develop mitochondrial markers, beside the fact that they sequenced the mt genome, since mt genes are known to be highly conserved, with a well known low rate of mutation? Are they better markers than plastidic one for the purpose of wood genotyping?

Why developing SCAR markers instead of SNP detecting technologies that need less DNA and are better suited to genotype large samples?

In addition, it is not clear to me whether the aim is to define markers specific to Fs as suggested in the introduction or only at the genus level.

M & M need more details: How many individuals/species were tested (only 4?). It seems to me that it’s hard at this level to assess the specificity for Fs as seems to acknowledge the authors. It is written that the sample is as representative as possible, but we need details to know what was really genotyped (location origin and so on). The essay of marker 3-Fagus-matR was tested on a larger sample that should be once again described. I think also that the 2 other markers should follow the same process and should be part of the present study.

L’email a bien été copié   L’email a bien été copié

Reviewer 3 Report

The development of the SNP and CAPS markers is interesting and should prove useful to the forest product industry. This portion of the work was well done and described.

Unfortunately, the first part of the paper describing the genome itself is uninformed by the literature on plant mitochondrial genomes and results in errors and omissions.

Important things to know about these genomes include that they are large and variable in size (many papers by Bendich, none of them cited), and that there are frequent rearrangements but low mutation rates (numerous papers not cited, starting with Palmer and Herbon, 1988). The sequences generally have large repeats that mediate recombinational isomerization within a species (many papers not cited, starting with Palmer and Herbon again) and numerous non-tandem repeats of 50bp and up that mediate recombination sometimes within a species, and likely the rearrangements seen between species (Wynn and Christensen, 2019 and Kozik et al, 2019, and many references cited therein).

Most importantly, plant mitochondrial genomes are NOT circular. Although they can usually map as circles, usually just one, this is due to their complex structure of multiple overlapping linear molecules. Therefore the statement in the manuscript (line 102) that the genome “is of circular structure” is incorrect and must be removed. The authors should carefully read Kozik et al, 2019, (the title of the paper even includes the statement “One ring does not rule them all”) to find out more about genome structures (there is even an appendix or supplement that includes a large number of references to angiosperm genomes and their structures). The paper also provides a way of accurately assembling these peculiar molecules, using long reads, and finding the multiple paths through the sequence.

The authors do not provide any evidence for circularity of the Fagus mitochondrial genome molecules, so they cannot state that they are circular. The assembly may best fit on a circular map, but that is not evidence for a circular genomic molecule. This is an important distinction that must be appreciated. It is important for the field that the authors do not perpetuate this misconception.

There is no information about non-tandem repeats or alternate paths through the genome. Almost all angiosperm mitochondrial genomes have large (>1kb) non-tandem repeats (SIlene conica has one that is 75kb) and they are recombinationally active (see above). It is not clear whether there is one in F. sylvatica because the genome sequence was not available to reviewers, and the authors do not report finding one (nor looking for one). One relative, Betula, has a large non-tandem repeat of about 500bp, and Quercus sequence has one larger than 1000bp, typical for an angiosperm. It is not credible that Fagus lacks a large non-tandem repeat. If the authors cannot find one, they need to check that it wasn’t discarded during assembly, which happens. The non-tandem repeats of 50bp and up can easily be found as described by Wynn and Christensen, 2019.

Closely related species have mitochondrial genomes that are related to each other usually by a series of inversions and occasionally duplications and deletions (Palmer and Herbon, 1988, see also Kozik et al, 2019). It would be very instructive to compare the gene order of Fagus to Quercus and Betula. Comparison to an outgroup may not be possible, but could be attempted. The rearrangements found might also be useful in taxonomy and forensics.

Round 2

Reviewer 2 Report

The revised version is greatly improved, new analyses are presented, with a better description of the new mt genome. Plant material used for CAPS validation is better detailed and the explanation of the strategy for mt marker development at the genus level is clear now.  The authors followed scrupulously former recommendations, and modified substantially their manuscript which is now very informative and more convincing. As a consequence, I strongly recommend the present paper for publication.

Just a minor question:   Lines 136-142, how do the authors explain the higher coverage of plastid derived sequences integrated in the mt genome? Do reads come from mt and plastid DNAs?

Author Response

Answer to Reviewer 2 (related to your minor question):

Thank you very much for the evaluation of the revised version of our manuscript. Yes, you are right. The higher coverage is due to an unspecific mapping of cpDNA-derived reads in addition to mtDNA-derived reads to these regions. Probably, much more reads coming from cpDNA than from mtDNA mapped to these regions. In genome skimming studies, it had been shown that the cpDNA depth is often higher than the mtDNA depth in total DNA from plant tissues (e.g., see Table 1 in Straub et al., 2012, American Journal of Botany 99: 349–364). The origin of a mapped read (mt or cp) could only be identified based on plastid-specific SNPs (SNPs not present in the mtDNA).

To make it hopefully clearer in the manuscript we made the following editions:

>Lines 140-142 after revision 1:

These three regions were also featured by increased coverage (compared to all other regions) when mapping the Illumina reads from DNA-sequencing of FASYL_29 to the related mitochondrial genome sequence (data not shown).

>Lines 140-142 after revision 2:

These three regions were also featured by increased coverage (compared to all other regions) when mapping all available trimmed Illumina reads from DNA-sequencing of FASYL_29 to the related mitochondrial genome sequence (data not shown). The increased coverage is due to an unspecific mapping of chloroplast DNA-derived reads in addition to mtDNA-derived reads to these regions.

Reviewer 3 Report

This paper is much improved compared to the original version. It is acceptable.

Author Response

Answer to Reviewer 3:

Thank you very much for the evaluation of the revised version of our manuscript.